# Comparison of Antigen Tests and qPCR in Rapid Diagnostics of Infections Caused by SARS-CoV-2 Virus

**DOI:** 10.3390/v14010017

**Published:** 2021-12-23

**Authors:** Adrianna Klajmon, Aldona Olechowska-Jarząb, Dominika Salamon, Agnieszka Sroka-Oleksiak, Monika Brzychczy-Włoch, Tomasz Gosiewski

**Affiliations:** 1Department of Molecular Biology, John Paul II Hospital, ul. Prądnicka 80, 31-202 Kraków, Poland; adrianna.klajmon@gmail.com; 2Department of Microbiology, John Paul II Hospital, ul. Prądnicka 80, 31-202 Kraków, Poland; aolechow@szpitaljp2.krakow.pl; 3Department of Molecular Medical Microbiology, Chair of Microbiology, Faculty of Medicine, Jagiellonian University Medical College, ul. Czysta 18, 31-121 Krakow, Poland; dominika.salamon@uj.edu.pl (D.S.); agnieszka.sroka@uj.edu.pl (A.S.-O.); m.brzychczy-wloch@uj.edu.pl (M.B.-W.)

**Keywords:** SARS-CoV-2, COVID-19, laboratory diagnostics, antigen test, NAAT, qPCR

## Abstract

Diagnostics of the coronavirus disease 2019 (COVID-19) using molecular techniques from the collected respiratory swab specimens requires well-equipped laboratory and qualified personnel, also it needs several hours of waiting for results and is expensive. Antigen tests appear to be faster and cheaper but their sensitivity and specificity are debatable. The aim of this study was to compare a selected antigen test with quantitative polymerase chain reaction (qPCR) tests results. Nasopharyngeal swabs were collected from 192 patients with COVID-19 symptoms. All samples were tested using Vitassay qPCR SARS-CoV-2 kit and the Humasis COVID-19 Ag Test (MedSun) antigen immunochromatographic test simultaneously. Ultimately, 189 samples were tested; 3 samples were excluded due to errors in taking swabs. The qPCR and antigen test results were as follows: 47 positive and 142 negative, and 45 positive and 144 negative, respectively. Calculated sensitivity of 91.5% and specificity of 98.6% for the antigen test shows differences which are not statistically significant in comparison to qPCR. Our study showed that effectiveness of the antigen tests in rapid laboratory diagnostics is high enough to be an alternative and support for nucleic acid amplification tests (NAAT) in the virus replication phase in the course of COVID-19.

## 1. Introduction

Coronaviruses (CoVs) are positive-stranded, non-segmented RNA enveloped viruses with an RNA of 27-32 kb and belong to the family *Coronaviridae*. To date, seven species of coronaviruses are known to infect humans: 4 species of the genus *Alphacoronavirus*—human CoV-NL63 [HCoV-NL63] [1], HCoV-OC43 [2,3], HCoV-229E [2,3], and HCoV-HKU1 [4]—and 3 species of *Betacoronavirus* genus—severe acute respiratory syndrome coronavirus (SARS-CoV), severe acute respiratory syndrome coronavirus-2 (SARS-CoV-2) and Middle East respiratory syndrome coronavirus (MERS-CoV) [5].

The species of the genus *Alphacoronavirus* cause cold symptoms, while the species of *Betacoronavirus* genus are zoonotic and are associated with severe acute respiratory tract infections. The epidemic caused by SARS-CoV emerged in 2002 in Guangdong province, China. SARS-CoV originated in horseshoe bats from where it was transmitted to the animal world and then to humans by direct animal-human contact [6]. Further transmission of this virus occurred from person to person. In 2012, Middle East respiratory syndrome coronavirus (MERS-CoV) emerged in Saudi Arabia and Jordan. The largest human-to-human transmission has occurred in Riyadh and Jeddah in 2014 and in South Korea in 2015. Dromedary camels are considered an animal source of MERS-CoV to human infection [7]. Major symptoms of MERS-CoV are low grade fever, chills, headache, nonproductive cough, dyspnea, myalgia. Severe cases reported hyperkalemia with associated ventricular tachycardia, disseminated intravascular coagulation, pericarditis, kidney damage, multi-organ failure and death [7,8,9].

In December 2019 in Hubei province, China, the severe acute respiratory syndrome coronavirus-2 (SARS-CoV-2) emerged [10]. This virus causes the coronavirus disease 2019 (COVID-19) with common symptoms of infection including fever, cough, shortness of breath, breathing difficulties, and in many cases loss of taste or smell. In severe cases, infections can cause pneumonia, severe acute respiratory distress syndrome, and kidney damage, as well as death. The less specific symptoms include muscle or body aches, headache, throat pain, congestion or runny nose and gastrointestinal symptoms—diarrhea, and less often vomiting [11,12]. Symptoms of the disease may appear 2–14 days after exposure to SARS-CoV-2 virions [13]. 

In accordance with the recommendations of the World Health Organization, which raised a global warning and announced the need for testing COVID-19-suspected patients, for the diagnostics procedures and identification of SARS-CoV-2 virus, samples from the upper respiratory tract (nasopharyngeal swab, oropharyngeal swab, nasopharyngeal aspirate, nasal wash or pharyngeal swab) and/or (if the patient is hospitalized) samples from the lower respiratory tract (bronchoalveolar lavage, endotracheal aspirates or expectorated sputum) are used [14]. According to Centers for Disease Control and Prevention (CDC) recommendations, diagnostic testing is intended to identify current infection in subjects and is conducted when a person has signs or symptoms consistent with COVID-19, or when an unvaccinated person has no symptoms of COVID-19 but was/or is suspected of exposure to SARS-CoV-2. Subjects with symptoms consistent with COVID-19 should be tested using nucleic acid amplification tests (NAAT) or antigen tests independently of vaccination status. A negative antigen test in subjects with signs or symptoms of COVID-19 should be confirmed by NAAT due to greater sensitivity [15].

The quantitative polymerase chain reaction (qPCR) belongs to the NAAT and it is the gold standard in the field of molecular laboratory diagnostics of SARS-CoV-2 infection [16]. However, the qPCR is expensive and requires access to a laboratory and equipment, and a time of about 3 h to obtain the results. Alternatives are sought, such as fast, cheap antigen tests, with minimal sample preparation requirements, but their sensitivity and specificity are debatable. There are a large number of the antigen test manufacturers on the market with a small amount of clinical trials. At the same time, there is a lack of globally accepted quality standards for antigen tests, which results in the presence of inaccurate tests on the market. The Foundation for Innovative Diagnostics (FIND) has maintained online lists of antigen and other molecular-based tests for SARS-CoV-2 detection (FIND 2020). At the time of writing (8 December 2021), FIND listed 15 rapid antigen tests that are in development, which indicates that there is still a large need for access to this type of test [17]. The aim of this study was to compare the antigen assays Humasis COVID-19 Ag Test kit (Humasis Co., Ltd., Gunpo-si, Korea) with qPCR tests results.

## 2. Materials and Methods

### 2.1. Patients

A total of 192 symptomatic patients were subjected to testing procedures from November 2020 to April 2021 in order to diagnose SARS-CoV-2 infection at the John Paul II Hospital in Cracow, Poland. All the research reported in this manuscript was performed in accordance with the ethical standards in the 1964 Declaration of Helsinki and was approved by the Jagiellonian University Ethical Committee (no. 1072.6120.132.2021) on 16 June 2021. All the participants provided their written informed consent.

The main and inclusion criterion were symptoms suggesting COVID-19 indicating the initial stage of infection with SARS-CoV-2 virus. All samples meeting the above criteria were included in the laboratory diagnostic process.

### 2.2. Laboratory Tests

The samples were subjected to the diagnostic process immediately after they were delivered to the laboratory. All tested samples were collected from the nasopharynx and both nostrils were swabbed for each test. Samples for PCR diagnostics were placed in the NUCLISWAB^®^ standard transport medium (Innovative Biotechnology Organization Ltd. (TiBO), Istanbul, Turkey). Samples dedicated for antigen diagnostics were collected on recommended flocked swabs and placed in a tube that is part of the Humasis COVID-19 Ag Test kit (Lot No. COVGC0017, Exp. 29.04.2022) (Humasis Co., Ltd., Gunpo-si, Korea).

### 2.3. Microscopic Analysis

In order to verify the correctness of collecting the virus material, we have introduced an internal microscopic verification procedure in our laboratory. To ensure that the swabs contained upper respiratory epithelial cells, after the diagnostic procedures safranin-stained microscopic slides were prepared. The samples were analyzed under microscope BX63 (Olympus Corporation, Corporate Pkwy, Center Valley, PA, USA) at a magnification of 600×.

### 2.4. qPCR

All samples were tested using molecular techniques (qPCR) and the Humasis COVID-19 Ag Test (Humasis Co., Ltd., Gunpo-si, Korea) antigen immunochromatographic test simultaneously. The swab intended to RNA isolation was vortexed and applied in a volume of 300 μL to the pre-filled 96-well plate with reagents. Then 10 µL of Proteinase K enzyme was added to the sample. The RNA isolation was performed automatically using the TANBead^®^ Nucleic Acid Extraction Kit (automated nucleic acid extraction instrument: Maelstrom 4800 isolator; Taiwan Advanced Nanotech Inc., Taoyuan City, Taiwan). The qPCR results were obtained on a GeneProof croBEE Real-Time PCR System thermocycler (GeneProof a.s., Brno, Czech Republic) using the Vitassay qPCR SARS-CoV-2 kit (Vitassay Healthcare S.L.U., Huesca, Spain) according to manufacturer’s protocol. This assay detected specific fragments of two SARS-CoV-2 genes (ORF1ab, N) and was equipped with a positive, negative and internal amplification control (IC). Results for each 96-well plate were assessed after approximately 3.5 h of extraction and qPCR process. The qPCR results were the cycle threshold (Ct) values, i.e., the point at which the amplification signal was read wasthe reaction cycle in which the product growth reached the established Ct ≤ 35. To increase the diagnostic reliability of qPCR results, we decreased the RFU boundary to ≤35 instead of the ≤38 suggested by manufacturer. 

### 2.5. SARS-CoV-2 Virus Antigen Detection 

The detection of antigens of the SARS-CoV-2 virus nucleocapsid protein and the RBD domain of the SARS-CoV-2 fusion protein (S protein) was performed using Humasis COVID-19 Ag Test kit (Humasis Co., Ltd., Gunpo-si, Korea) according to the manufacturer’s protocol by immersing a swab from the patient in the extraction buffer, stirring the buffer five times, and squeezing the edges of the tube to squeeze as much material as possible from the swab. The sample in a volume of 3 drops (90–100 μL) was applied to the test cassette inside which a nitrocellulose membrane coated with anti-SARS-CoV-2 antibodies was placed.

The result was read after 15 min. The test contained an internal control in the form of the C control line, which was visible within the test cassette with each correctly performed test. After starting the test, the results for all 30 samples were assessed after approximately 0.5 h.

### 2.6. Statistical Analysis

Quantitative variables were presented as mean values (M) ± standard deviation (SD), qualitative data as numbers and percentages. All parameters of the antigen diagnostic test were calculated with reference to the gold standard (qPCR). However, the reliability of the antigen diagnostic test was assessed using the Cohen’s Kappa coefficient along with the Z test for checking the significance of Cohen’s Kappa coefficient. In the analysis, *p* < 0.05 was considered statistically significant. The statistical analysis was performed with the use of the STATISTICA 13.3 statistical package (StatSoft Europe, Hamburg, Germany).

## 3. Results

A total of 192 patients suspected of infection by SARS-CoV-2 were studied. Of them, three samples were excluded due to errors in taking swabs. Among 189 included samples (61.1% men, 39.9% women) the mean age was 64.7 years (±13.9).

### 3.1. Microscopic Analysis 

The presence of epithelial cells (visible ciliary cell) was confirmed in all swabs, which guaranteed correct collection of the material (Figure 1A,B). In control material, collected incorrectly, the epithelial cells were not detected, instead epidermal cells from the nasal vestibule were visible, enabling reliable SARS-CoV-2 detection (Figure 1C,D).

### 3.2. Humasis COVID-19 Ag Test and Vitassay qPCR SARS-CoV-2 Kit Results

The results of the determinations are summarized in Table 1. Among the qualified samples, non-diagnostic results were not obtained in the PCR tests. There were also no defective samples (without a control line) in the determinations performed with the indicated antigen test. 

### 3.3. Positive qPCR Results

Of the 189 samples analyzed, 47 (24.9%) were qPCR positive. Ct values of the positive samples are summarized in Table 2. Mean Ct values of the samples were 21.8 ± 5.8 and 24 ± 5.2 for the ORF1ab gene and N gene respectively. In our study, four patients with a negative antigen test response had a Ct > 31 for the ORF1ab gene and a Ct > 24 for the N gene. 

### 3.4. Negative qPCR Results 

One hundred and forty-two (75.1%) samples were qPCR negative. Of them, 140 obtained negative results from the antigen test, and 2 obtained positive results. Due to the negative result of the PCR test, the Ct values were not reported. 

### 3.5. Positive Results in the Antigen Test and Negative Results in the qPCR Test 

There were two samples (1.05%) with positive antigen test results being negative in the qPCR test.

The comparison of the two tests (Table 1), taking into account the positive results, suggests that the cut-off sensitivity level of the monitored Humasis COVID-19 Ag Test kit (Lot No. COVGC0017, Exp. 29.04.2022) may be around Ct = 31 for the ORF1ab gene. 

With reference to the obtained results, the calculated diagnostic values of the Humasis COVID-19 Ag Test kit were presented in Table 3. 

Both diagnostic tests were compatible in 90.03% of the cases (Cohen’s Kappa ratio= 0.9003) (*p* < 0.001).

## 4. Discussion

Our goal was to assess the effectiveness of a new antigen test on the market in the rapid diagnosis of infections caused by the SARS-CoV-2 virus. This study was performed to show that the Humasis Ag COVID-19 test is effective and can be a support for qPCR testing of SARS-CoV-2 in the early stages of infection, with suspicious clinical symptoms observed that may suggest COVID-19 disease. According to WHO guidelines patients suspected of COVID-19 have higher priority than asymptomatic patients in terms of testing strategy. Furthermore, it is said that only asymptomatic individuals can be tested if they are at high risk of infection—for example, health care workers—particularly in settings where NAAT testing abilities are limited [18]. Turcato et al. indicate that the antigen test in asymptomatic patients is much less sensitive (63.1%) compared to symptomatic patients (89.8%) [19]. Mitchell et al. report that antigen testing should be performed on patients within 5 days of onset of symptoms (test sensitivity 87.8%) compared to asymptomatic patients (sensitivity 33.3%) [20].

Sungnak et al. (2020) consider that goblet cells and ciliary cells show the highest expression of the angiotensin-converting enzyme 2 protein necessary for the virus to enter the cell and determining the SARS-CoV-2 replication [21]. With regard to this knowledge, our study suggests that correct swab collection is essential in early stages of viral infection and transmission (Figure 1).

In our study, four samples with negative antigen test results and positive in qPCR were observed only at high Ct-values, i.e., Ct > 31. This can be explained by the fact that the amount of antigens available in collected samples was below the sensitivity level of the test. It is even more interesting considering the fact that we chose more restrictive requirements for Ct (Ct < 35) values in our study than suggested by the manufacturer. 

Low SARS-CoV-2 RNA concentrations occur at early stages of infection, before the viral replication phase, or in a late stage of infection when replication has decreased, resulting in a low antigen content, below the sensitivity level of antigen tests. Patients with negative antigen results in the early infection stages may be infectious, which is important in terms of virus transmission [22]. It seems that a good solution is to verify negative antigen tests with the qPCR test, and such is the recommendation in Poland (“... negative result of the test requires verification if the clinical picture or significant epidemiological indications suggest COVID-19, because a negative result of the antigen test does not rule out infection”) [23]. In our study, we only investigated symptomatic patients because they most likely contribute to onward infection transmission. Obtaining two samples negative in qPCR and positive in the antigen test may suggest a possible contamination at the stage of collecting the material or its transport. Moreover, possible cross-reactions cannot be ruled out. The rapid mutation rate of the virus can also lead to a false negative result due to alteration of the nucleotide sequence [24]. Calculated sensitivity of 91.5% and specificity of 98.6% shows differences which are not statistically significant in comparison to qPCR (the gold standard) and is high enough to be a support for NAAT, which is in line with Chaimaio et al. (2020) and Porte et al. (2020), as well as suggesting that antigen detection tests can be used as a screening assay [25,26,27]. It has been shown that in some specimens SARS-CoV-2 RNA can be detected up to six weeks after recovery from COVID-19 [28]. These observations are in line with recommendations for a symptom-based strategy for ending isolation of people who are not considered to be infectious. In such situations, the antigen tests can replace the qPCR tests and may allow early termination of patient isolation. Moreover, no laboratory equipment is needed, nor is specialized molecular knowledge necessary to carry out antigen tests, and they can be performed as a form of self-test in household conditions (albeit in accordance with legal regulations and recommendations of national health care agencies) or in public places. Moreover, results similar to qPCR can be obtained in a shorter time. These features allow for rapid identification of infected patients, thus preventing further virus transmission, especially in places where people gather, such as emergency or waiting rooms, which is in accordance with Krüttge et al. (2020), despite the fact that the group received inferior results of the antigen test used [29]. Based on available data, the cost of the antigen test is approximately 2/3 lower than nucleic acid amplification tests [16]. Adopting antigen testing in symptomatic patient diagnostics can speed up the SARS-CoV-2 detection process, anticipating the next wave of infections and reducing the burden on virology labs that have been overwhelmed during the recent COVID-19 pandemic.

### Study Limitations

This study has several limitations. First, the studied group was relatively small and the associations between antigen test and qPCR test results need to be studied in a larger cohort of patients. Second, in our study we only tested the symptomatic patients; to estimate the exact effectiveness of the test it is necessary to conduct the studies also on asymptomatic patients, which is the subject of our next research. 

## Figures and Tables

**Figure 1 viruses-14-00017-f001:**
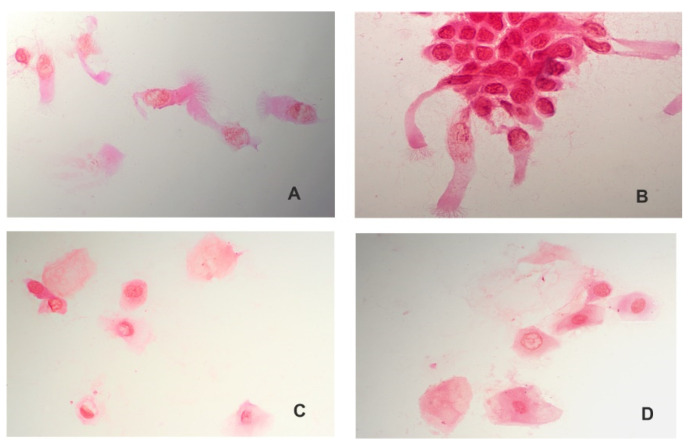
Swabs collected correctly from nasopharynx—visible ciliary cells (**A**,**B**). Swabs collected incorrectly—epidermal cells from the nasal vestibule are visible (**C**,**D**). Magnification 600x.

**Table 1 viruses-14-00017-t001:** Summary of the results obtained by the qPCR method and the Humasis COVID-19 Ag Test.

Test Result		qPCR Test		
		Positive	Negative	Total
Humasis COVID-19 Ag Test	Positive	43	2	45
	Negative	4	140	144
	Total	47	142	189

**Table 2 viruses-14-00017-t002:** Summary of the positive-sample Ct values obtained by the qPCR method and the Humasis COVID-19 Ag Test.

qPCR Test	ORF 1ab Gene Ct Values	N Gene Ct Values	Humasis COVID-19 Ag Test
Positive	29.95	37.87	Positive
Positive	15.16	21.04	Positive
Positive	11.49	15.56	Positive
Positive	18.97	23.89	Positive
Positive	11.46	15.55	Positive
Positive	19.60	17.90	Positive
Positive	26.62	23.42	Positive
Positive	24.55	39.46	Positive
Positive	22.97	23.93	Positive
Positive	16.65	24.20	Positive
Positive	31.72	32.95	Positive
Positive	18.47	23.72	Positive
Positive	21.44	22.39	Positive
Positive	20.84	21.07	Positive
Positive	15.64	28.48	Positive
Positive	18.47	23.72	Positive
Positive	21.22	21.89	Positive
Positive	17.72	22.90	Positive
Positive	23.93	23.73	Positive
Positive	16.16	19.84	Positive
Positive	22.19	21.38	Positive
Positive	24.08	24.27	Positive
Positive	30.46	24.29	Positive
Positive	28.04	31.07	Positive
Positive	33.37	31.03	Positive
Positive	30.42	24.90	Positive
Positive	28.31	28.11	Positive
Positive	24.89	25.65	Positive
Positive	24.05	26.11	Positive
Positive	14.35	16.64	Positive
Positive	18.57	24.37	Positive
Positive	22.89	24.54	Positive
Positive	11.95	11.56	Positive
Positive	11.12	17.83	Positive
Positive	22.24	24.65	Positive
Positive	29.76	28.20	Positive
Positive	28.78	26.06	Positive
Positive	17.72	22.90	Positive
Positive	18.37	17.48	Positive
Positive	22.12	23.76	Positive
Positive	19.81	24.99	Positive
Positive	27.25	24.35	Positive
Positive	23.10	24.27	Positive
Positive	31.80	23.90	Negative
Positive	32.81	31.02	Negative
Positive	35.10	33.21	Negative
Positive	34.20	33.61	Negative

**Table 3 viruses-14-00017-t003:** Diagnostic values of the Humasis COVID-19 Ag Test.

	Value	95% CI
**Sensitivity**	91.49%	79.62–97.63%
**Specificity**	97.90%	93.99–99.57%
**PPV**	93.48%	82.10–98.63%
**NPV**	97.22%	93.04–99.24%
**LR+**	43.61	14.19–134.07
**LR-**	0.09	0.03–0.22
**Accuracy**	96.32%	92.56–98.51%

Abbreviations: CI—confidence interval, PPV—positive predictive value, NPV—negative predictive value, LR—likelihood ratio.

## Data Availability

Not applicable.

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
