# Peer review of "Comparison of Antigen Tests and qPCR in Rapid Diagnostics of Infections Caused by SARS-CoV-2 Virus"

_viruses, 2021, doi:10.3390/v14010017_

Round 1

Reviewer 1 Report

My concerns have been addressed

Author Response

Thank you for your review.

Reviewer 2 Report

The authors adequately made the required changes

Author Response

Thank you for your review.

Reviewer 3 Report

The authors approach a very actual theme. The use of rapid tests for detecting the SARS-Cov 2 virus is currently very important for the clinical practice.

The introduction should be improved with more informations on the available antigen tests and their advantages in comparison with PCR based techniques. 

The authors compared the results for the rapid tests with the results of qPCR tests for symptomatic patients. On my opinion it is more important to establish the sensitivity and specificity of the rapid tests for asymptomatic patients as more and more those tests are used in schools as a screening method to identify the children that are COVID 19 positive. The authors mentioned that "antigen tests are dedicated to screening 257
symptomatic patients" which is not exactly correct, as many countries are using those tests also as a screening method for asymptomatic persons. Another limitation of the study is the fact that is was used only for patients the initial stage of infection, it would be important to assess the limitation of the antigen tests in patients at different time points in order to have a clear understanding of the tests performances. 

The Discussion section should be more informative as regard the performances of other antigen tests developed for detecting the SARS-Cov 2. 

Round 2

Reviewer 3 Report

It should be better to have the asymptomatic patients testing presented in the same paper.

This manuscript is a resubmission of an earlier submission. The following is a list of the peer review reports and author responses from that submission.

Round 1

Reviewer 1 Report

To evaluate effectiveness of the single COVID-19 antigen test kit, authors tested 192 nasopharyngeal samples from patients with COVID-19 symptoms in comparison with qPCR test results. Calculated sensitiviry and specificity of antigen test was 91.5% and 98.6%, respectively. Authors suggested that antigen test could be an alternatives for COVID-19 diagnosis by NAAT. 

However, authors evaluated samples from symptomatic patient only, and the number of samples were not enough to get a solid conclusion. Authors could describe the limitation of research design, because they did not use samples from non-symptomatic patients.  Moreover, false negative results in antigen tests could undermine efforts at containment and potentially lead to outbreaks in vulnerable patients. There has been many reports evaluating various antigen test, and the sensitivity varied between studies. It could be helpful to refer various reports and discuss how the antigen tests could be suitable for alternative and support for NAAT.  

In addition, there was no information of Ct values of each samples, except false-negative 4 samples >31. This cut-off value is different from other reports ranged from 23 to 30. It is needed to describe Ct values of each samples and its distribution.     

In Materials and Methods, all test samples were collected form the nasopharynx and treated simultaneously by two different buffer, transport medium for qPCR and extraction buffer for antigen test. Collect both sides and use one side each? Describe correct sampling and processing methods.    

Reviewer 2 Report

In my opinion the article is available for publication

Reviewer 3 Report

Comparing the antigen test and PCR test gave expected results with a small number of samples giving different results in each of the tests. However the study did not convince me that its better to use a slightly cheaper/faster test that has the chance of giving inaccurate results when these results may need confirming anyway. The authors also discussed the fact that several antigen tests exist with no real oversight. It would have been interesting to test additional antigen kits. 

Minor corrections in grammar required. In some places the text was cumbersome. For example the title could be shortened "Comparison of antigen tests and qPCR in rapid diagnostics of infections caused by SARS-CoV-2".